# PipeFusion: Patch-level Pipeline Parallelism for Diffusion Transformers Inference

**Jiarui Fang** [*†]
ByteDance
fangjiarui123@gmail.com

**Jinzhe Pan**[*†]
HUST
eigensystem@hust.edu.cn

**Aoyu Li**[†]
ByteDance
aoyuli2000@outlook.com

**Xibo Sun**
Tencent
xibosun@tencent.com

**Jiannan Wang**[†]
The University of Hong Kong
steaunk@connect.hku.hk

## Abstract

This paper presents PipeFusion, an innovative parallel methodology to tackle the high latency issues associated with generating high-resolution images using diffusion transformers (DiTs) models. PipeFusion partitions images into patches and the model layers across multiple GPUs. It employs a patch-level pipeline parallel strategy to orchestrate communication and computation efficiently. By capitalizing on the high similarity between inputs from successive diffusion steps, PipeFusion reuses one-step stale feature maps to provide context for the current pipeline step. This approach notably reduces communication costs compared to existing DiTs inference parallelism, including tensor parallel, sequence parallel and DistriFusion. PipeFusion enhances memory efficiency through parameter distribution across devices, ideal for large DiTs like Flux.1. Experimental results demonstrate that PipeFusion achieves state-of-the-art performance on $8 \times$L40 PCIe GPUs for Pixart, Stable-Diffusion 3, and Flux.1 models. Our Source code is available at `https://github.com/xdit-project/xDiT`.

## 1   Introduction

In recent years, the capabilities of AI-generated content (AIGC) have rapidly advanced, with diffusion models emerging as a leading generative technique in image [1–4] and video synthesis [5, 6]. The model architecture of diffusion models is undergoing a significant transformation. Traditionally dominated by U-Net [7] architectures, these models are now transitioning to Diffusion Transformers (DiTs) [4]. The context length of diffusion models is also increasing, which is crucial for handling long videos and high-resolution images.

DiTs suffer from quadratic latency growth in long-sequence generation due to attention computation. Given that a single GPU cannot satisfy the latency requirements for practical applications, it becomes necessary to parallelize the DiTs inference for a single image across multiple computational devices. However, tensor parallelism (TP) [8], commonly used in serving Large Language Models (LLMs), is inefficient for DiTs due to their large activation volumes. In such cases, the communication costs often outweigh the benefits of parallel computation. To address the challenges posed by long sequences, sequence parallelism (SP) [9, 10] is applied for the parallel inference of DiTs, which partitions the input image into patches and needs data exchange across different devices in the attention operations.

---

[*]Equal contribution.

[†]This work was completed when Jiarui Fang, Jinzhe Pan, Aoyu Li, and Jiannan Wang were with Tencent.

39th Conference on Neural Information Processing Systems (NeurIPS 2025).

Recent studies have highlighted the presence of *input temporal redundancy* in Diffusion Transformers (DiTs) during inference [11–15, 6]. This phenomenon manifests as significant similarity in both inputs and activations across consecutive diffusion time steps, presenting opportunities for optimizing parallel computation approaches. DistriFusion [16] represents a sequence parallelism method that exploits this characteristic in U-Net-based diffusion models. The approach combines fresh local activations with stale activations from preceding steps during attention and convolution operations, effectively overlapping communication costs within individual diffusion timestep computations. However, applying DistriFusion to DiTs reveals substantial memory inefficiencies, as the method requires maintaining full spatial dimensions of attention keys (K) and values (V) across all layers, resulting in memory costs that scale linearly with sequence length.

To more effectively utilize input temporal redundancy, we propose **PipeFusion**, a novel patch-level pipeline parallelism. PipeFusion partitions both the DiTs layers and input image patches across GPUs, enabling pipelined parallel processing by leveraging temporal redundancy to reuse stale activations from previous timesteps. Firstly, PipeFusion substantially reduces inter-device communication costs compared to existing DiTs' parallel methods. It only transfers the input of the initial layer and the output of the final layer on each device, whereas other parallel approaches, i.e. sequence and tensor parallel, require transmitting activations for every DiT layer. Secondly, PipeFusion offers lower memory consumption by distributing model parameters across multiple devices. This makes it particularly suitable for DiTs with large model parameters, such as the 12B Flux.1 model [1]. In contrast to DistriFusion, PipeFusion reduces the memory buffers required for attention K, V to $1/N$ ($N$ is the parallel degree). Finally, PipeFusion, by utilizing fresh K and V values over a longer period, achieves higher image generation accuracy compared to DistriFusion. Experimental results show that PipeFusion achieves the lowest DiTs inference latency on 8×L40 PCIe-connected GPUs compared to sequence parallelism, DistriFusion, and tensor parallelism.

## 2 Background & Related Works

**Diffusion Models and Diffusion Transformers (DiTs)**: Diffusion models generate high-quality images using a noise-prediction deep neural network (DNN) denoted by $\epsilon_\theta$. The process begins with pure Gaussian noise $x_T \sim \mathcal{N}(0, I)$ and iteratively denoises it through multiple steps to produce the final image $x_0$, with $T$ being the total number of diffusion time steps. As shown in Figure 1, at each step $t$, given the noisy image $x_t$, the model $\epsilon_\theta$ takes $x_t$, $t$, and an additional condition $c$ (e.g., text, image) as inputs to predict the noise $\epsilon_t$ within $x_t$. The previous image $x_{t-1}$ is obtained using:

$$x_{t-1} = \text{Update}(x_t, t, \epsilon_t), \quad \epsilon_t = \epsilon_\theta(x_t, t, c). \tag{1}$$

The function Update is specific to the sampler (e.g., DDIM [17], DPM [18]) and involves element-wise operations. Finally, $x_0$ is decoded from the Latent Space to the Pixel Space using a Variational Autoencoder (VAE) [19]. The main contributor to inference latency is the forward propagation through $\epsilon_\theta$. The architecture of $\epsilon_\theta$ is transitioning from U-Net [7] to Diffusion Transformers (DiTs) [20–22], driven by the scaling law that shows improved performance with increased model parameters and training data. Unlike U-Nets that use convolutional layers to capture spatial hierarchies, DiTs segment the input into latent patches and use the transformer's self-attention mechanism [23] to model relationships within and across these patches. The input noisy latent representation is decomposed into patches, embedded into tokens, and fed into a series of DiT blocks, which generally include Multi-Head Self-Attention, Layer Norm, and Feedforward Networks.

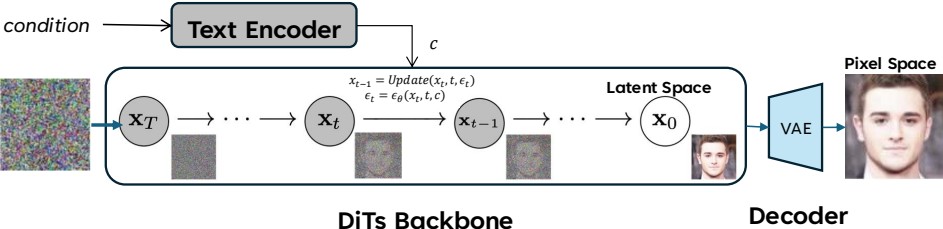

Figure 1: Workflow of DiTs inference.

**Input Temporal Redundancy:** Diffusion Model entails the iterative prediction of noise from the input image or video. Recent research has highlighted the concept of *input temporal redundancy*,

which refers to the similarity observed in both the inputs and activations across successive diffusion timesteps [11, 12]. A recent study [13] further investigates the distribution of this similarity across different layers and timesteps. Based on the redundancy, a branch of research caches the activation values and reuses them in the subsequent diffusion timesteps to prune computation. For example, in the context of the U-Net architecture, DeepCache updates the low-level features while reusing the high-level ones from cache [12]. Additionally, TGATE caches the cross-attention output once it converges along the diffusion process [12]. By contrast, in the domain of DiT models, Δ-DiT proposes to cache the rear DiT blocks in the early sampling stages and the front DiT blocks in the later stages [24]. PAB [15] exploits the U-shaped attention pattern to mitigate temporal redundancy through a pyramid-style broadcasting approach. Finally, DiTFastAttn [14] identifies three types of redundancies, such as spatial redundancy, temporal redundancy, and conditional redundancy, and presents an attention compression method to speed up generation.

## 3  DiT Parallel Methods

In this section, we delve into the application of tensor parallelism (TP), sequence parallelism (SP), and an asynchronous SP leveraging input temporal redundancy, namely DistriFusion, for parallel inference in DiTs. Following this, we introduce a novel patch-level pipeline parallel approach named PipeFusion that markedly improves communication efficiency compared to existing techniques.

In the remainder of this article, the following notations are used: $p$ denotes the sequence length, which corresponds to the number of pixels in the latent space. $hs$ represents the hidden size of the model. $L$ stands for the number of network layers. $N$ is the number of computing devices.

### 3.1  Tensor Parallelism & Sequence Parallelism

Tensor parallelism (TP) partitions the linear layers of DiTs column-wise and row-wise. This approach requires two all-reduce operations per layer, incurring a communication overhead of $4O(p \times hs)$ per layer (Table 1). The communication cost scales linearly with the sequence length, leading to poor scalability, especially for large models such as the 0.6B-parameter PixArt model (Figure 8). Additionally, TP poses implementation challenges for non-standard architectures like MM-DiT in Flux/SD3.

Sequence parallelism (SP) partitions images into patches, with each device processing a patch as input. Since attention mechanisms require global interactions, devices must communicate partial query (Q), key (K), and value (V) between each other. As shown in Figure 2, two notable SP implementations are DeepSpeed-Ulysses [9] (SP-Ulysses) and Ring-Attention [10] (SP-Ring). SP-Ulysses uses All2All communication to switch between sequence and hidden-size partitioning during parallel attention, while SP-Ring implements distributed Flash Attention via point-to-point K/V subblock transfers. It is worth noting that SP-Ulysses and SP-Ring can be combined to form USP (Unified Sequence Parallel) [25], which is often more efficient than using either method alone.

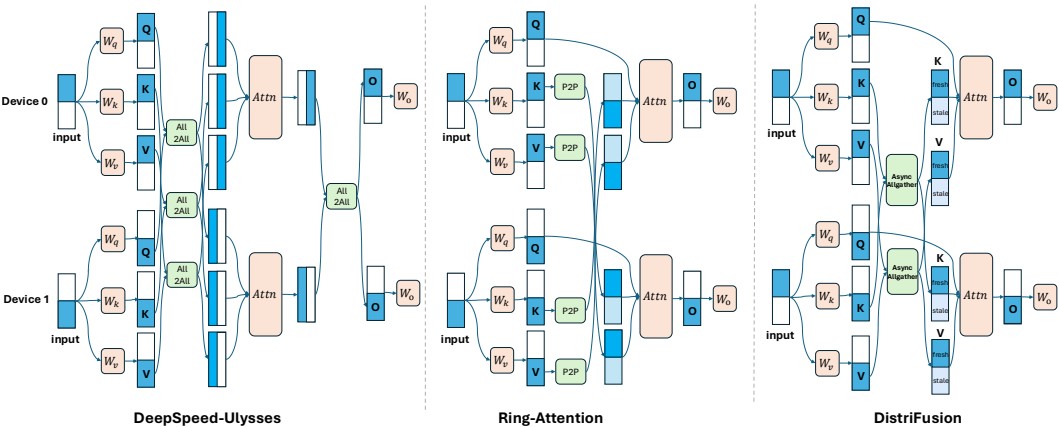

Figure 2: Comparison of DistriFusion with Sequence Parallelism methods (DeepSpeed-Ulysses and Ring-Attention) for attention layers.

## 3.2 DistriFusion: Asynchronous Sequence Parallelism

DistriFusion is an asynchronous sequence parallelism method that leverages temporal redundancy in the input data. It uses slightly outdated (stale) activations from the previous timestep, rather than requiring fresh activations at each step. Despite this, DistriFusion maintains image generation accuracy without any perceptible loss in quality.

While DistriFusion was initially applied to U-Net-based SDXL models [26], it can be extended to DiT models as well. As illustrated in Figure 2, DistriFusion also operates by partitioning along the sequence dimension. The key distinction, however, lies in its use of asynchronous all-gather operations to collect K and V activations from remote devices. Due to the asynchronous nature of this communication, the activations for the current diffusion step are not immediately available; instead, they become accessible during the next diffusion step.

DistriFusion exploits this by utilizing a fraction $\frac{N-1}{N}$ of the K and V activations from timestep $T + 1$, combined with a fraction $\frac{1}{N}$ of the local K and V from diffusion timestep $T$. This combination is used to compute the attention operation with the local queries at diffusion timestep $T$. The communication of K and V for timestep $T$ is intentionally overlapped with the network's forward computation at timestep $T$, thus hiding communication overhead during computation.

However, DistriFusion achieves this communication-computation overlap at the cost of increased memory usage. Each computational device is required to maintain communication buffers that store the complete spatial shape of the K and V activations, which amounts to $AL$ in total. Consequently, the memory cost of DistriFusion does not scale down with the addition of computational devices.

## 3.3 PipeFusion: Patch-level Pipeline Parallel

PipeFusion simultaneously partitions the input sequence and the layers of the DiTs backbone as shown at the top of Figure 3. PipeFusion partitions the DiTs model along the direction of data flow. Therefore, each partition contains a set of consecutive layers and is deployed on a GPU. PipeFusion also partitions the input image into $M$ non-overlapping patches so that each GPU processes the computation for one patch with its assigned layers in parallel, and the diffusion routine runs efficiently in a pipelined manner. Pipelining requires synchronizations between devices, which would be efficient if the DiT workload is evenly distributed across each GPU. The even partition is easy to achieve as a DiT contains number of identical transformer blocks.

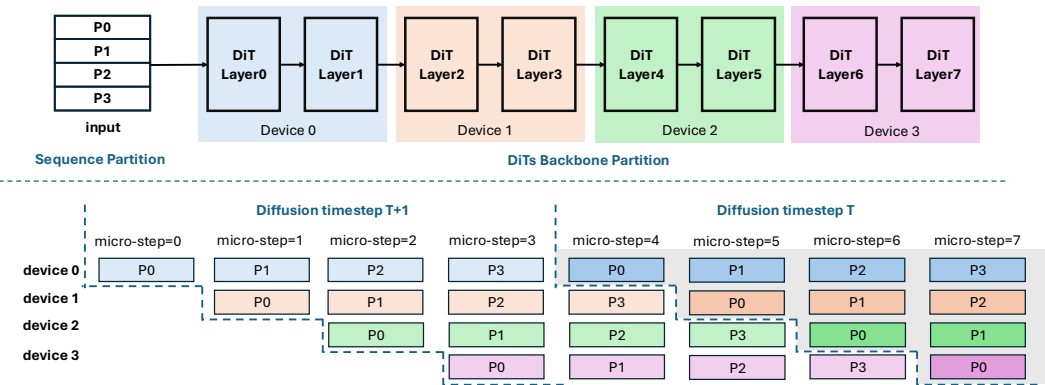

Figure 3: Above: partitioning strategy for Input and DiTs backbone network. Below: Workflow of the PipeFusion as patch-level pipelined parallelism.

Suppose we are currently at diffusion timestep $T$, the previous timestep is $T + 1$ as the diffusion process goes along the reversed order of timesteps. The example in Figure 3 demonstrates the pipeline workflow with $N = 4$ and $M = 4$, where the activation values of patches at timestep $T$ are highlighted. Due to the input temporal redundancy, a device does not need to wait for the receiving of full spatial shape activations for the current timestep $T$ to start the computation of its own stage. Instead, it employs the stale activations from the previous timestep to provide context for the current computation. For instance, in micro-step 5, the activations for Patchs P0 and P1 are from timestep $T$, while others are from timestep $T + 1$. In this way, each device maintains the full activations and

there is no waiting time after the pipeline is initialized. Bubbles only exist at the beginning of the pipeline. Suppose the total number of diffusion timesteps is $S$, then the effective computation ratio of the pipeline is $\frac{M \cdot S}{M \cdot S + N - 1}$. As $S$ is commonly chosen to be a large number for high-quality image generation, the effective computation ratio is high. For example, when $M = N = 4$ and $S = 50$, the ratio is equal to 98.5%. As long as the patch number $M$ is greater than the parallel degree $N$, there will be no bubble generation except for the initial startup overhead. We recommend setting $M = N$. We have analyzed the impact of $M$ on the final performance in Appendix Sec. B.

Moreover, in PipeFusion, a device sends micro-step patch activations to the subsequent device via asynchronous P2P, enabling the overlap between communication and computation. For example, at micro-step 4, device 0 receives P1's activation from diffusion timestep $T$ while concurrently computing the activation values for P0 at timestep $T$. Similarly, at the next micro-step, the transmission of P0 to device 1 can be hidden by the computation of P1 on the device.

Similar to DistriFusion, before executing the pipeline, we usually conduct several diffusion iterations synchronously, called warmup steps. During the warmup phase, patches are processed sequentially, resulting in low efficiency. Though the warmup steps cannot be executed in a pipelined manner, the workflow is relatively small compared to the entire diffusion process, and thus, the impact on performance is negligible. We have analyzed the impact of warmup and propose the corresponding solutions in Appendix Sec. B.

PipeFusion theoretically outperforms DistriFusion in terms of generation quality, considering the area of fresh activation. As shown in Figure 4, within a single diffusion timestep, PipeFusion continuously increases the area of fresh activation as the pipeline micro-steps progress from diffusion timestep 4 to 8. In contrast, throughout the entire diffusion process, DistriFusion constantly maintains one patch of fresh area out of the total $M$ patches. Therefore, the results of PipeFusion are closer to the original diffusion method compared to DistriFusion.

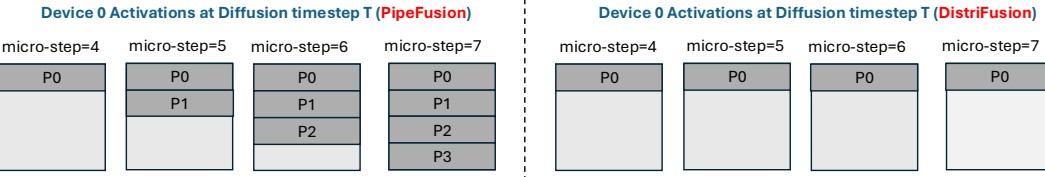

Figure 4: The fresh part of activations during diffusion timestep $T$ of Figure 3. The dark gray represents fresh data and the light gray represents stable data.

The exploitation of input temporal redundancy requires a short warmup phase, since the behavior of diffusion synthesis changes qualitatively over the course of denoising. Accordingly, DistriFusion performs several warmup steps with standard synchronous patch parallelism before switching to its asynchronous mode. For PipeFusion, warmup steps cannot be pipelined and therefore introduce bubbles. However, because the number of warmup steps is typically very small compared to the total diffusion iterations, their overall impact on efficiency is limited. We analyze this effect in Sec. B.

PipeFusion represents a fundamental departure from prior pipeline parallelization methods, such as GPipe [27] and TeraPipe [28]. Unlike GPipe, which splits along the batch dimension and depends on high concurrency across multiple requests, PipeFusion partitions along the sequence dimension to deliver efficient pipelined execution for a single image. Unlike TeraPipe, which is designed for causal attention in language models, PipeFusion is tailored to the full-attention mechanism of DiTs and exploits temporal redundancy to simultaneously improve efficiency and preserve generation quality.

### 3.4 Comparison Between Different Parallelisms

Compared with existing parallel solutions, PipeFusion demonstrates superior efficiency in communication and memory usage. Firstly, PipeFusion transmits between computational devices only the activations that serve as inputs and outputs for a series of consecutive transformer layers belonging to a stage, significantly reducing the communication bandwidth requirement in comparison to other methods that transmit the KV activations for all the $L$ layers. In particular, the total communication cost for PipeFusion is $2O(p \times hs)$, not associated with the number of layers $L$. Secondly, each device in PipeFusion stores only $\frac{1}{N}$ of the parameters belonging to its specific layers. Since the usage of

Table 1: Comparison between different DiT parallel methods on a single diffusion timestep. Overlap denotes the overlap between communication and computation.

| Method | Communication | | Memory Cost | |
|---|---|---|---|---|
| | Cost | Overlap | Model | KV Activations |
| Tensor Parallel | $4O(p \times hs)L$ | ✘ | $\frac{1}{N}P$ | $\frac{1}{N}KV$ |
| DistriFusion | $2O(p \times hs)L$ | ✔ | $P$ | $(KV)L$ |
| SP-Ring | $2O(p \times hs)L$ | ✔ | $P$ | $\frac{1}{N}KV$ |
| SP-Ulysses | $\frac{4}{N}O(p \times hs)L$ | ✘ | $P$ | $\frac{1}{N}KV$ |
| PipeFusion | $2O(p \times hs)$ | ✔ | $\frac{1}{N}P$ | $\frac{1}{N}(KV)L$ |

stale KV for attention computation requires each device to maintain the full spatial KV for the $\frac{L}{N}$ layers on the device, the overhead is significantly smaller than that of DistriFusion and diminishes as the number of devices increases.

In the following, we summarize the theoretical cost of existing methods in Table 1. The communication cost is calculated by the product of the number of elements to transfer with an *algorithm bandwidth (algobw)* factor related to communication type [3]. For collective algorithms of AllReduce, AllGather and AllToAll, the corresponding algobw factors are $2\frac{n-1}{n}$, $\frac{n-1}{n}$, and 1. In the table, we approximate the term $O(\frac{n-1}{n})$ to $O(1)$ for simplicity.

Among all methods, PipeFusion has the lowest communication cost, as long as $N < 2L$, which is easy to satisfy as the number of network layers $L$ is typically quite large, e.g., $L = 38$ in Stable-Diffusion-3. In addition, PipeFusion overlaps communication and computation. Ulysses Seq Parallel exhibits a decreasing communication cost with increasing $N$, outperforming the remaining three methods. However, its communication cannot be hidden by computation. Ring Seq Parallel and DistriFusion have similar communication costs and overlapping behaviors. The distinction lies in the scope of overlapping: computation in Ring Seq Parallel overlaps within the attention module, whereas that in DistriFusion overlaps throughout the entire forward pass.

To analyze the memory costs, we denote $P$ as the total number of model parameters. In PipeFusion and tensor parallelism, the memory cost decreases as more GPUs are utilized, which is a nice property with the rapid growth of the DiT model size. Both PipeFusion and DistriFusion maintain a KV buffer for each transformer layer leading to significant activation memory overhead, especially for long sequences. The KV buffer in PipeFusion decreases as the number of devices $N$ increases, whereas DistriFusion does not exhibit such a reduction.

## 4  Experiments

In our experimental setup, we deployed our trials on an 8×L40-48GB (PCIe Gen4x16) cluster to evaluate three prominent DiT models: **Pixart** [21, 3]: Its backbone is a transform model that encompasses 0.6B (billion) parameters. Pixart draws its architectural foundation from the original DiTs framework, with a pivotal enhancement: the integration of cross-attention modules designed to incorporate text conditions into the model's processing. **Stable-Diffusion-3** [2] (SD3-medium): which leverages a Multimodal (MM)-DiT and its backbone transformer model contains 2B parameters. As the largest open-source variant available, SD3-medium employs a 20-step FlowMatchEulerDiscreteScheduler. **Flux.1-dev** [1]: Its backbone transformer model contains 12B parameters and features an enhanced MM-DiT design. Flux.1-dev also relies on a 28-step FlowMatchEulerDiscreteScheduler.

### 4.1  Performance Results

This section compares the performance of different parallel methods on 8×L40 GPUs, connected by PCIe Gen4. This section only presents the latency of the DiT except for the latency of the VAE decoding, since VAE module is the same across different parallel methods. Since the Pixart and SD3 can utilize Classifier-Free Guidance (CFG) [29], we additionally implemented the CFG parallel. For an input prompt, CFG parallel individually computes the forward tasks for both unconditional

---

[3]https://github.com/NVIDIA/nccl-tests/blob/master/doc/PERFORMANCE.md

guidance and text guidance. It leverages inter-image parallel to perform these two tasks, collecting the results after each diffusion step is completed. For PipeFusion, we select the best latency performance by searching the patch number $M$ from 2, 4, 8, 16, 32. All figures show the average latency of 5 runs. We employ a default warmup step of 1 for both DistriFusion and PipeFusion. The software stack utilized includes PyTorch 2.4.1, CUDA Runtime 12.1.105, and diffusers 0.30.3.

### 4.1.1 Latency Evaluation

**Pixart:** Figure 5 shows the inference latency of Pixart on image generation tasks with resolution as 1024px (1024×1024), 2048px (2048×2048) and 4096px (4096×4096) on 8×L40.

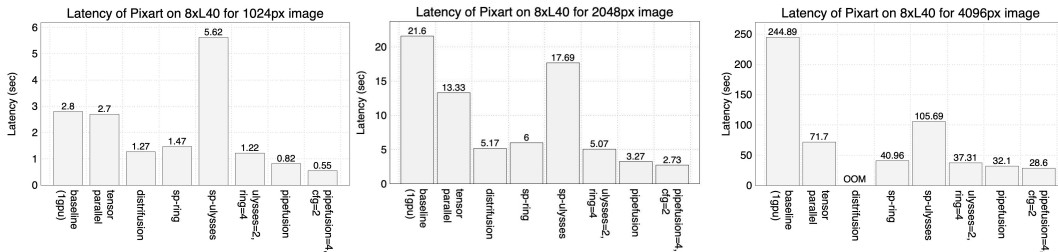

Figure 5: Latency on Pixart of various parallel approaches on two image generation tasks with the 20-Step DPMSolverMultistepScheduler.

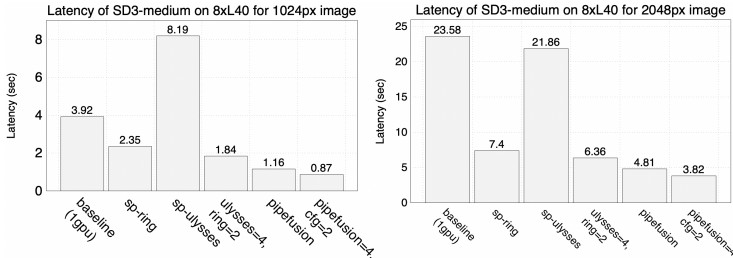

Figure 6: Latency on SD3-medium of various parallel approaches on two image generation tasks with the 20-Step FlowMatchEulerDiscrete Scheduler.

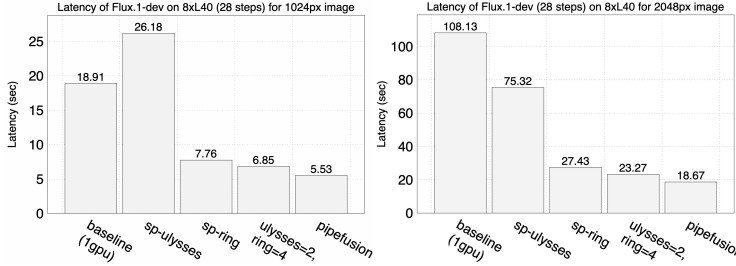

Figure 7: Latency on Flux.1-dev of various parallel approaches on three image generation tasks with the 28-Step FlowMatchEulerDiscrete Scheduler.

PipeFusion exhibits the lowest latency among parallel methods across all three tasks. ***PipeFusion is 1.48×, 1.55×, and 1.16× faster than the second best parallel approaches on the 1024px, 2048px and 4096px tasks.***

The highest latency parallelism is always SP-Ulysses, primarily due to the bandwidth bottleneck between CPU sockets by conducting All2All on 8 GPUs via PCIe. SP-Ring exhibits significantly lower latency compared to SP-Ulysses. An efficiency method USP [25] can further improve SP performance by hybridizing SP-Ulysses and SP-Ring. It views the process group as a 2D mesh where the columns are SP-Ring groups and the rows are SP-Ulysses groups. USP parallelizes the attention head dimension in SP-Ulysses groups and parallelizes the sequence dimension in SP-Ring groups.

USP yields a lower latency than SP-Ring, as illustrated by the ulysses=2 and ring=4 columns in the figure.

Additionally, tensor parallel also shows high latency, which aligns with our analysis in section 3.4. DistriFusion exhibits slightly higher latency compared to the USP (ulysses=2 and ring=4), and it encounters out-of-memory (OOM) issues on the 4096px task.

Using the CFG parallel, the 8 GPUs are divided into two groups, each performing PipeFusion independently, which can further reduce latency. This approach (cfg=2, pipefusion=4 in the figure) achieves a latency reduction of 5.09x, 7.91x, and 8.59x compared to the baseline (1 GPU). PipeFusion achieves perfect scalability in tasks 2048px and 4096px.

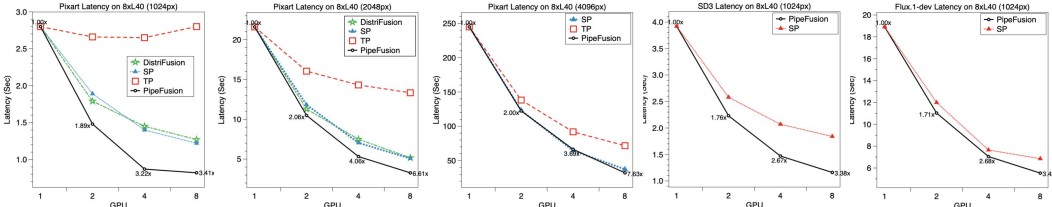

Figure 8: Scalability Analysis of Three DiTs on 8×L40: Leftmost figures depict Pixart performance on 1024px, 2048px, and 4096px images. The second from the right shows SD3 performance on 1024px. The rightmost figure illustrates Flux.1-dev performance on 1024px. The speedup to 1 GPU baseline is labeled on the points of pipefusion curves.

**Stable Diffusion 3:** Figure 6 shows the inference latency of SD3-medium on image generation tasks with resolution as 1024px, 2048px on 8×L40. The model is unable to generate 4096px images due to positional encoding limitations. Since MM-DiT applied in SD3 is a non-standard transformer layer, arranging column-wise and row-wise weight partitioning poses significant challenges, and we did not implement tensor parallelism. Additionally, due to potential OOM issues with DistriFusion and inferior performance compared with USP, we did not implement it for SD3.

PipeFusion outperforms the best sequence parallelism in these two cases, both of which are also the USP with ulysses=4, ring=2, *achieving speedups of 1.57× and 1.30×.* When combined with CFG parallel, the pipefusion=4, cfg=2 scheme on 8 GPUs achieves speedups of 3.11× and 8.16× compared to the baseline (1 GPU).

**Flux.1-dev:** Figure 7 illustrates the inference latency of SD3-medium on image generation tasks with resolutions of 1024px and 2048px, respectively, using 8×L40. Flux.1 employs a similar MM-DiT architecture to SD3 but extends the network depth to 57 layers.

In both scenarios, PipeFusion surpasses the best sequence parallelism, which is also the USP with ulysses=4 and ring=2, *achieving speedups of 1.23× and 1.25×.* PipeFusion offers 3.42× and 5.79× speedup compared to the baseline (1 GPU) for the two cases.

**Scalability**: We evaluated the scalability of inference for three models by scaling from 1 to 2, 4, and 8 GPUs. Figure 8 illustrates the scalability of different parallel approaches. PipeFusion and DistriFusion both apply a single warmup iteration, while for SP we adopt the USP, which searches the optimal combination of Ulysses and Ring degrees. The results show that PipeFusion consistently achieves the best scalability across all five scenarios. Given that PixArt has the smallest model size, tensor parallelization unsurprisingly exhibits the worst scalability.

At 1024px and 2048px resolutions, PipeFusion substantially outperforms both SP and DistriFusion. Even at 4096px, where DistriFusion encounters out-of-memory (OOM) issues and thus provides no valid data, PipeFusion continues to deliver strong performance: on 8×L40 GPUs, it achieves 32.1s end-to-end latency versus SP's best 37.3s, corresponding to a 1.16× speedup. While this speedup appears smaller than at lower resolutions, PipeFusion's communication efficiency remains a clear advantage. Specifically, PipeFusion reduces communication share to only 4.6% of total latency at 4096px, compared with 17.9% for SP.

Here, the communication share is defined as

$$\text{CommShare} = \frac{T_{\text{E2E, 8GPU}} - \frac{T_{\text{single}}}{8}}{T_{\text{E2E, 8GPU}}}, \tag{2}$$

which subtracts 1/8 of the single-GPU latency from the observed 8-GPU latency. For example,

$$\frac{32.1 - 244.89/8}{32.1} \approx 4.6\% \quad \text{(PipeFusion)}, \qquad \frac{37.3 - 244.89/8}{37.3} \approx 17.9\% \quad \text{(SP)}.$$

In absolute terms, SP's communication cost is 6.69s, whereas PipeFusion lowers it to 1.49s — a 78% reduction. This shows that even in the compute-bound regime, PipeFusion robustly suppresses communication overhead.

### 4.1.2 Memory Efficiency

We collect the memory usage excluding the VAE usage, which involves the computation of the text-encoder and DiTs backbones, as shown inAs shown in Figure 1. The figure includes memory usage for "parameters", which encompasses both the text encoder and transformers, as well as for "activations", which includes activations and temporary buffers. Note that SP-Ulysses, SP-Ring and USP exhibit similar memory consumption, denoted as SP in the figure.

The maximum memory usage of different parallel approaches for Pixart is depicted in Figure 9a. The Pixart model consists of a 0.6B parameter DiT backbone (2.3 GB on disk) and T5-based text encoders (18GB on disk). So the memory consumption of parameters is dominated by the text encoder. DistriFusion maintains full spatial shape K, V tensors for each layer, resulting in an increase in memory consumption as the image resolutions grow. In contrast, PipeFusion only needs to store $1/N$ of the full spatial shape K, V tensors. Consequently, on $8\times$GPUs, its memory footprint is comparable to that of Tensor Parallel and Sequence Parallel. Remarkably, even for tasks with an 8192px resolution, PipeFusion exhibits the lowest memory consumption.

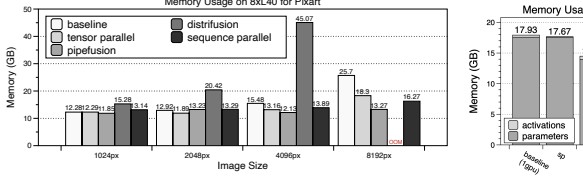
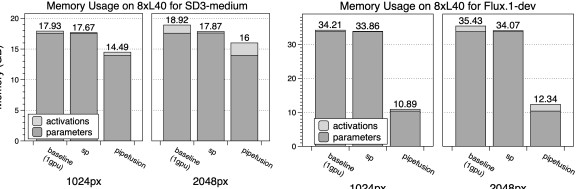

(a) Max GPU Memory Usage of Various Approaches for Pixart Image Generation Tasks Across Four Resolutions.

(b) Max GPU Memory Usage on 8 GPUs for SD3-medium and Flux.1 on 1024px and 2048px image generation tasks.

Figure 9: Comparison of GPU memory usage across different models and resolutions.

The maximum memory usage of sequence parallel and PipeFusion for SD3-medium and Flux.1-dev is depicted in Figure 9b. The SD3-medium consists of a 2B parameter DiTs backbone (7.8 GB on disk) and text encoders of nearly 19GB on disk. The Flux.1-dev consists of a 12B parameter DiTs backbone (23 GB on disk) and text encoders of nearly 9.1GB on disk. The memory cost of PipeFusion is also significantly less than that of sequence parallelism. For the 12B Flux.1-dev, PipeFusion significantly reduces the memory footprint of model parameters. However, PipeFusion increases the memory usage of activations due to the consumption of the KV Buffer. Nevertheless, this increase is relatively minor compared to the overall model parameter memory usage. ***The overall memory usage of PipeFusion is 32% and 36% of SP on 1024px and 2048px cases using Flux.1***.

### 4.2 Quality Results

Table 2 presents the Fréchet Inception Distance (FID) [30] scores for PipeFusion, DistriFusion, and the Original model, evaluated on the Pixart and Flux.1 datasets. A lower FID score is indicative of superior performance in terms of image quality. The results clearly demonstrate that PipeFusion consistently outperforms DistriFusion across identical device configurations, achieving significantly lower FID scores. This finding is in direct alignment with the theoretical analysis presented in Sec. 3.3, where PipeFusion is shown to leverage fresher activations while effectively reducing the impact of stale ones, thereby enhancing overall performance.

Additionally, we provide a comprehensive visual comparison in Fig. 10 of Appendix A, where image samples generated by PipeFusion, DistriFusion, and the Original model are displayed. The images generated by PipeFusion are nearly indistinguishable from the original images to the human eye.

| Model | PixArt-XL-2-256-MS | | | | | | FLUX.1[dev] | | |
|---|---|---|---|---|---|---|---|---|---|
| **Method** | Original | DistriFusion | | PipeFusion | | | Original | PipeFusion | |
| **Device Number** | 1 | 4 | 8 | 2 | 4 | 8 | 1 | 4 | 8 |
| **FID** ↓  **w/ G.T.** | 22.78 | 30.69 | 41.81 | 23.60 | 25.23 | 28.46 | 25.03 | 24.17 | 25.97 |
| **w/ Orig.** | - | 8.41 | 19.81 | 1.67 | 3.11 | 6.09 | - | 4.01 | 5.93 |

Table 2: FID Scores for Parallel Methods on the Pixart and Flux.1. w/ G.T. means calculating the FID metrics with the ground-truth images. w/ Orig. means calculating the FID metrics with the images generated by the 1-Device original implementation.

# 5   Limitations & Discussion

While this paper has demonstrated PipeFusion's advantages over tensor parallelism and sequence parallelism, we acknowledge that it is not a universal solution. In fact, combining PipeFusion with existing approaches into a hybrid parallelization scheme is likely the most practical direction for deployment. Such a design enables efficient scaling of DiT inference across large cross-node configurations: sequence parallelism can be exploited within NVLink-connected, high-bandwidth settings, while PipeFusion is particularly effective across inter-node, low-bandwidth environments.

A key limitation of PipeFusion lies in its dependence on multiple diffusion steps. For models with only a handful of steps—such as distilled variants like the 4-step Flux.1-schnell [31] or one-step diffusion methods [32, 33]—temporal redundancy is minimal, making PipeFusion less effective. However, this does not pose a major constraint in practice, as state-of-the-art DiT models generally require a substantial number of steps to deliver competitive quality.

Another promising extension is the application of PipeFusion to video generation. Architecturally, PipeFusion is already compatible with multimodal DiTs (MM-DiTs) such as Flux.1 and Stable Diffusion 3, where cross-modal conditions are injected through unified attention. Recent video models (e.g., Wan2.1 [34], HunyuanVideo [35], CogVideo [36]) adopt similar MM-DiT backbones, suggesting that PipeFusion can be applied with minimal adaptation. From the tensor-shape perspective, both image and video DiTs operate on latent tensors of shape $(B, L, H)$, with video models flattening spatial and temporal dimensions into a unified sequence dimension. Because PipeFusion partitions along the sequence dimension, this structural difference does not affect applicability. From the performance perspective, video models are typically larger and involve longer sequence lengths, which amplifies communication overhead for SP. PipeFusion mitigates this overhead through parameter partitioning and pipelined execution, offering reduced memory footprint and stronger scalability. This suggests that video generation is not only a natural but also a highly favorable domain for applying PipeFusion.

# 6   Conclusion

We introduced PipeFusion, a patch-level pipeline parallelism strategy tailored for Diffusion Transformers. By reusing temporally redundant activations and overlapping communication with computation, PipeFusion reduces both bandwidth demands and memory footprint while preserving image quality. Experiments on Pixart, Stable Diffusion 3, and Flux.1 demonstrate consistent latency improvements over strong baselines, achieving up to 1.5× faster inference on commodity multi-GPU clusters.

Beyond efficiency gains, PipeFusion highlights a broader systems insight: temporal redundancy in diffusion processes can be systematically exploited to design new forms of parallelism. While our study focused on single-node, multi-GPU inference, the approach naturally composes with existing sequence or tensor parallelism, offering a path toward scalable deployment on large clusters. We hope this work inspires further exploration of hybrid parallel strategies that close the gap between rapidly growing generative models and the practical requirements of serving them at scale.

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

## NeurIPS Paper Checklist

1. **Claims**

   Question: Do the main claims made in the abstract and introduction accurately reflect the paper's contributions and scope?

   Answer: [Yes]

   Justification: The paper clearly states its contribution of proposing PipeFusion, a novel patch-level pipeline parallelism method for efficient inference of diffusion transformer models. It highlights the key ideas of leveraging input temporal redundancy and reducing communication and memory costs compared to existing parallel methods. The claims are supported by the experimental results demonstrating state-of-the-art performance on multiple models and configurations.

   Guidelines:

   - The answer NA means that the abstract and introduction do not include the claims made in the paper.
   - The abstract and/or introduction should clearly state the claims made, including the contributions made in the paper and important assumptions and limitations. A No or NA answer to this question will not be perceived well by the reviewers.
   - The claims made should match theoretical and experimental results, and reflect how much the results can be expected to generalize to other settings.
   - It is fine to include aspirational goals as motivation as long as it is clear that these goals are not attained by the paper.

2. **Limitations**

   Question: Does the paper discuss the limitations of the work performed by the authors?

   Answer: [Yes]

   Justification: The paper discusses limitations in Section 5, noting that PipeFusion is not well-suited for models with a limited number of diffusion steps, such as distilled models or one-step diffusion methods. It also mentions that while PipeFusion reduces communication and memory costs, it may face challenges in scenarios with extremely long sequences where communication overhead becomes less significant.

   Guidelines:

   - The answer NA means that the paper has no limitation while the answer No means that the paper has limitations, but those are not discussed in the paper.
   - The authors are encouraged to create a separate "Limitations" section in their paper.
   - The paper should point out any strong assumptions and how robust the results are to violations of these assumptions (e.g., independence assumptions, noiseless settings, model well-specification, asymptotic approximations only holding locally). The authors should reflect on how these assumptions might be violated in practice and what the implications would be.

- The authors should reflect on the scope of the claims made, e.g., if the approach was only tested on a few datasets or with a few runs. In general, empirical results often depend on implicit assumptions, which should be articulated.
- The authors should reflect on the factors that influence the performance of the approach. For example, a facial recognition algorithm may perform poorly when image resolution is low or images are taken in low lighting. Or a speech-to-text system might not be used reliably to provide closed captions for online lectures because it fails to handle technical jargon.
- The authors should discuss the computational efficiency of the proposed algorithms and how they scale with dataset size.
- If applicable, the authors should discuss possible limitations of their approach to address problems of privacy and fairness.
- While the authors might fear that complete honesty about limitations might be used by reviewers as grounds for rejection, a worse outcome might be that reviewers discover limitations that aren't acknowledged in the paper. The authors should use their best judgment and recognize that individual actions in favor of transparency play an important role in developing norms that preserve the integrity of the community. Reviewers will be specifically instructed to not penalize honesty concerning limitations.

3. **Theory assumptions and proofs**

Question: For each theoretical result, does the paper provide the full set of assumptions and a complete (and correct) proof?

Answer: [NA]

Justification: The paper focuses on proposing and evaluating the PipeFusion method through experimental results, and does not include formal theoretical results, theorems, or proofs.

Guidelines:

- The answer NA means that the paper does not include theoretical results.
- All the theorems, formulas, and proofs in the paper should be numbered and cross-referenced.
- All assumptions should be clearly stated or referenced in the statement of any theorems.
- The proofs can either appear in the main paper or the supplemental material, but if they appear in the supplemental material, the authors are encouraged to provide a short proof sketch to provide intuition.
- Inversely, any informal proof provided in the core of the paper should be complemented by formal proofs provided in appendix or supplemental material.
- Theorems and Lemmas that the proof relies upon should be properly referenced.

4. **Experimental result reproducibility**

Question: Does the paper fully disclose all the information needed to reproduce the main experimental results of the paper to the extent that it affects the main claims and/or conclusions of the paper (regardless of whether the code and data are provided or not)?

Answer: [Yes]

Justification: The paper provides sufficient details to reproduce the main experimental results. It describes the experimental setup, including the hardware configuration (8×L40 PCIe GPUs), software stack (PyTorch, CUDA Runtime, diffusers), and the models used (Pixart, Stable-Diffusion 3, Flux.1). It also explains the methodology for implementing PipeFusion and other parallel approaches, and provides results for various image resolutions and configurations. While the code and data are not explicitly provided, the detailed descriptions allow for replication of the experiments.

Guidelines:

- The answer NA means that the paper does not include experiments.
- If the paper includes experiments, a No answer to this question will not be perceived well by the reviewers: Making the paper reproducible is important, regardless of whether the code and data are provided or not.

- If the contribution is a dataset and/or model, the authors should describe the steps taken to make their results reproducible or verifiable.
- Depending on the contribution, reproducibility can be accomplished in various ways. For example, if the contribution is a novel architecture, describing the architecture fully might suffice, or if the contribution is a specific model and empirical evaluation, it may be necessary to either make it possible for others to replicate the model with the same dataset, or provide access to the model. In general. releasing code and data is often one good way to accomplish this, but reproducibility can also be provided via detailed instructions for how to replicate the results, access to a hosted model (e.g., in the case of a large language model), releasing of a model checkpoint, or other means that are appropriate to the research performed.
- While NeurIPS does not require releasing code, the conference does require all submissions to provide some reasonable avenue for reproducibility, which may depend on the nature of the contribution. For example
  - (a) If the contribution is primarily a new algorithm, the paper should make it clear how to reproduce that algorithm.
  - (b) If the contribution is primarily a new model architecture, the paper should describe the architecture clearly and fully.
  - (c) If the contribution is a new model (e.g., a large language model), then there should either be a way to access this model for reproducing the results or a way to reproduce the model (e.g., with an open-source dataset or instructions for how to construct the dataset).
  - (d) We recognize that reproducibility may be tricky in some cases, in which case authors are welcome to describe the particular way they provide for reproducibility. In the case of closed-source models, it may be that access to the model is limited in some way (e.g., to registered users), but it should be possible for other researchers to have some path to reproducing or verifying the results.

5. **Open access to data and code**

Question: Does the paper provide open access to the data and code, with sufficient instructions to faithfully reproduce the main experimental results, as described in supplemental material?

Answer: [Yes]

Justification: The paper cites the original sources for the models and datasets used, such as Pixart, Stable-Diffusion 3, and Flux.1. It also mentions the use of publicly available datasets like COCO Captions 2014 for evaluation. While it does not explicitly state the license for each asset, it is common practice in the field to assume that publicly available datasets and models are used under their respective standard licenses (e.g., CC-BY for COCO Captions). The paper does not release any new assets, so there is no need to provide licenses for derived assets.

Guidelines:

- The answer NA means that paper does not include experiments requiring code.
- Please see the NeurIPS code and data submission guidelines (`https://nips.cc/public/guides/CodeSubmissionPolicy`) for more details.
- While we encourage the release of code and data, we understand that this might not be possible, so "No" is an acceptable answer. Papers cannot be rejected simply for not including code, unless this is central to the contribution (e.g., for a new open-source benchmark).
- The instructions should contain the exact command and environment needed to run to reproduce the results. See the NeurIPS code and data submission guidelines (`https://nips.cc/public/guides/CodeSubmissionPolicy`) for more details.
- The authors should provide instructions on data access and preparation, including how to access the raw data, preprocessed data, intermediate data, and generated data, etc.
- The authors should provide scripts to reproduce all experimental results for the new proposed method and baselines. If only a subset of experiments are reproducible, they should state which ones are omitted from the script and why.

- At submission time, to preserve anonymity, the authors should release anonymized versions (if applicable).
- Providing as much information as possible in supplemental material (appended to the paper) is recommended, but including URLs to data and code is permitted.

6. **Experimental setting/details**

Question: Does the paper specify all the training and test details (e.g., data splits, hyper-parameters, how they were chosen, type of optimizer, etc.) necessary to understand the results?

Answer: [Yes]

Justification:The paper cites the original sources for the models and datasets used, such as Pixart, Stable-Diffusion 3, and Flux.1. It also mentions the use of publicly available datasets like COCO Captions 2014 for evaluation. While it does not explicitly state the license for each asset, it is common practice in the field to assume that publicly available datasets and models are used under their respective standard licenses (e.g., CC-BY for COCO Captions). The paper does not release any new assets, so there is no need to provide licenses for derived assets.

Guidelines:

- The answer NA means that the paper does not include experiments.
- The experimental setting should be presented in the core of the paper to a level of detail that is necessary to appreciate the results and make sense of them.
- The full details can be provided either with the code, in appendix, or as supplemental material.

7. **Experiment statistical significance**

Question: Does the paper report error bars suitably and correctly defined or other appropriate information about the statistical significance of the experiments?

Answer: [No]

Justification: The paper does not report error bars, confidence intervals, or other statistical significance tests for the experiments. It is because the results is stable among tests. It presents the average latency and FID scores for multiple runs but does not provide information on the variability or statistical significance of these results.

Guidelines:

- The answer NA means that the paper does not include experiments.
- The authors should answer "Yes" if the results are accompanied by error bars, confidence intervals, or statistical significance tests, at least for the experiments that support the main claims of the paper.
- The factors of variability that the error bars are capturing should be clearly stated (for example, train/test split, initialization, random drawing of some parameter, or overall run with given experimental conditions).
- The method for calculating the error bars should be explained (closed form formula, call to a library function, bootstrap, etc.)
- The assumptions made should be given (e.g., Normally distributed errors).
- It should be clear whether the error bar is the standard deviation or the standard error of the mean.
- It is OK to report 1-sigma error bars, but one should state it. The authors should preferably report a 2-sigma error bar than state that they have a 96% CI, if the hypothesis of Normality of errors is not verified.
- For asymmetric distributions, the authors should be careful not to show in tables or figures symmetric error bars that would yield results that are out of range (e.g. negative error rates).
- If error bars are reported in tables or plots, The authors should explain in the text how they were calculated and reference the corresponding figures or tables in the text.

8. **Experiments compute resources**

Question: For each experiment, does the paper provide sufficient information on the computer resources (type of compute workers, memory, time of execution) needed to reproduce the experiments?

Answer: [Yes]

Justification: The paper provides detailed information on the computer resources used for the experiments. It specifies the hardware configuration as 8×L40 PCIe GPUs with 48GB memory each and mentions the software stack including PyTorch 2.4.1 and CUDA Runtime 12.1.105. Additionally, it reports the average latency for multiple runs, which indicates the time of execution for each experiment.

Guidelines:

- The answer NA means that the paper does not include experiments.
- The paper should indicate the type of compute workers CPU or GPU, internal cluster, or cloud provider, including relevant memory and storage.
- The paper should provide the amount of compute required for each of the individual experimental runs as well as estimate the total compute.
- The paper should disclose whether the full research project required more compute than the experiments reported in the paper (e.g., preliminary or failed experiments that didn't make it into the paper).

9. **Code of ethics**

Question: Does the research conducted in the paper conform, in every respect, with the NeurIPS Code of Ethics https://neurips.cc/public/EthicsGuidelines?

Answer: [Yes]

Justification: The research described in the paper does not involve human subjects, sensitive data, or any other elements that would require special ethical considerations according to the NeurIPS Code of Ethics. The paper focuses on improving the efficiency of diffusion transformer models for image generation, which does not pose risks related to privacy, discrimination, security, or other ethical concerns outlined in the guidelines.

Guidelines:

- The answer NA means that the authors have not reviewed the NeurIPS Code of Ethics.
- If the authors answer No, they should explain the special circumstances that require a deviation from the Code of Ethics.
- The authors should make sure to preserve anonymity (e.g., if there is a special consideration due to laws or regulations in their jurisdiction).

10. **Broader impacts**

Question: Does the paper discuss both potential positive societal impacts and negative societal impacts of the work performed?

Answer: [No]

Justification: The paper does not discuss the broader societal impacts of the work performed. While it focuses on improving the efficiency of diffusion transformer models for image generation, it does not address potential positive or negative societal impacts. The research is foundational and does not directly tie to specific applications or deployments that could have significant societal consequences.

Guidelines:

- The answer NA means that there is no societal impact of the work performed.
- If the authors answer NA or No, they should explain why their work has no societal impact or why the paper does not address societal impact.
- Examples of negative societal impacts include potential malicious or unintended uses (e.g., disinformation, generating fake profiles, surveillance), fairness considerations (e.g., deployment of technologies that could make decisions that unfairly impact specific groups), privacy considerations, and security considerations.

- The conference expects that many papers will be foundational research and not tied to particular applications, let alone deployments. However, if there is a direct path to any negative applications, the authors should point it out. For example, it is legitimate to point out that an improvement in the quality of generative models could be used to generate deepfakes for disinformation. On the other hand, it is not needed to point out that a generic algorithm for optimizing neural networks could enable people to train models that generate Deepfakes faster.
- The authors should consider possible harms that could arise when the technology is being used as intended and functioning correctly, harms that could arise when the technology is being used as intended but gives incorrect results, and harms following from (intentional or unintentional) misuse of the technology.
- If there are negative societal impacts, the authors could also discuss possible mitigation strategies (e.g., gated release of models, providing defenses in addition to attacks, mechanisms for monitoring misuse, mechanisms to monitor how a system learns from feedback over time, improving the efficiency and accessibility of ML).

11. **Safeguards**

Question: Does the paper describe safeguards that have been put in place for responsible release of data or models that have a high risk for misuse (e.g., pretrained language models, image generators, or scraped datasets)?

Answer: [NA]

Justification: The paper does not release any datasets or models that pose a high risk for misuse. It focuses on the methodology and experimental results of the PipeFusion approach without distributing any models or datasets that could be misused.

Guidelines:

- The answer NA means that the paper poses no such risks.
- Released models that have a high risk for misuse or dual-use should be released with necessary safeguards to allow for controlled use of the model, for example by requiring that users adhere to usage guidelines or restrictions to access the model or implementing safety filters.
- Datasets that have been scraped from the Internet could pose safety risks. The authors should describe how they avoided releasing unsafe images.
- We recognize that providing effective safeguards is challenging, and many papers do not require this, but we encourage authors to take this into account and make a best faith effort.

12. **Licenses for existing assets**

Question: Are the creators or original owners of assets (e.g., code, data, models), used in the paper, properly credited and are the license and terms of use explicitly mentioned and properly respected?

Answer: [Yes]

Justification: The paper cites the original sources for the models and datasets used, such as Pixart, Stable-Diffusion 3, and Flux.1. It also mentions the use of publicly available datasets like COCO Captions 2014 for evaluation. While it does not explicitly state the license for each asset, it is common practice in the field to assume that publicly available datasets and models are used under their respective standard licenses (e.g., CC-BY for COCO Captions). The paper does not release any new assets, so there is no need to provide licenses for derived assets.

Guidelines:

- The answer NA means that the paper does not use existing assets.
- The authors should cite the original paper that produced the code package or dataset.
- The authors should state which version of the asset is used and, if possible, include a URL.
- The name of the license (e.g., CC-BY 4.0) should be included for each asset.
- For scraped data from a particular source (e.g., website), the copyright and terms of service of that source should be provided.

- If assets are released, the license, copyright information, and terms of use in the package should be provided. For popular datasets, `paperswithcode.com/datasets` has curated licenses for some datasets. Their licensing guide can help determine the license of a dataset.
- For existing datasets that are re-packaged, both the original license and the license of the derived asset (if it has changed) should be provided.
- If this information is not available online, the authors are encouraged to reach out to the asset's creators.

13. **New assets**

    Question: Are new assets introduced in the paper well documented and is the documentation provided alongside the assets?

    Answer: [NA]

    Justification: The paper does not introduce or release any new assets such as datasets, models, or code. Therefore, there is no need to document or provide documentation for new assets.

    Guidelines:

    - The answer NA means that the paper does not release new assets.
    - Researchers should communicate the details of the dataset/code/model as part of their submissions via structured templates. This includes details about training, license, limitations, etc.
    - The paper should discuss whether and how consent was obtained from people whose asset is used.
    - At submission time, remember to anonymize your assets (if applicable). You can either create an anonymized URL or include an anonymized zip file.

14. **Crowdsourcing and research with human subjects**

    Question: For crowdsourcing experiments and research with human subjects, does the paper include the full text of instructions given to participants and screenshots, if applicable, as well as details about compensation (if any)?

    Answer: [NA]

    Justification: The paper does not involve any crowdsourcing experiments or research with human subjects. It focuses on the technical aspects of parallelizing diffusion transformer models for image generation, which does not require human participation or data collection.

    Guidelines:

    - The answer NA means that the paper does not involve crowdsourcing nor research with human subjects.
    - Including this information in the supplemental material is fine, but if the main contribution of the paper involves human subjects, then as much detail as possible should be included in the main paper.
    - According to the NeurIPS Code of Ethics, workers involved in data collection, curation, or other labor should be paid at least the minimum wage in the country of the data collector.

15. **Institutional review board (IRB) approvals or equivalent for research with human subjects**

    Question: Does the paper describe potential risks incurred by study participants, whether such risks were disclosed to the subjects, and whether Institutional Review Board (IRB) approvals (or an equivalent approval/review based on the requirements of your country or institution) were obtained?

    Answer: [NA]

    Justification: The paper does not involve any research with human subjects, so IRB approvals or equivalent reviews are not applicable.

    Guidelines:

- The answer NA means that the paper does not involve crowdsourcing nor research with human subjects.
- Depending on the country in which research is conducted, IRB approval (or equivalent) may be required for any human subjects research. If you obtained IRB approval, you should clearly state this in the paper.
- We recognize that the procedures for this may vary significantly between institutions and locations, and we expect authors to adhere to the NeurIPS Code of Ethics and the guidelines for their institution.
- For initial submissions, do not include any information that would break anonymity (if applicable), such as the institution conducting the review.

16. **Declaration of LLM usage**

Question: Does the paper describe the usage of LLMs if it is an important, original, or non-standard component of the core methods in this research? Note that if the LLM is used only for writing, editing, or formatting purposes and does not impact the core methodology, scientific rigorousness, or originality of the research, declaration is not required.

Answer: [No]

Justification: The paper does not describe the usage of Large Language Models (LLMs) as an important, original, or non-standard component of the core methods. The research focuses on the development and evaluation of the PipeFusion method for parallelizing diffusion transformer models, which is a technical contribution unrelated to LLMs.

Guidelines:

- The answer NA means that the core method development in this research does not involve LLMs as any important, original, or non-standard components.
- Please refer to our LLM policy (https://neurips.cc/Conferences/2025/LLM) for what should or should not be described.

# A Image Quality

In Figure 10, we selected a few sample images generated using prompts from DistriFusion [16] to help readers visually assess generation quality. The images generated by PipeFusion are nearly indistinguishable from the original images to the human eye, across various configurations of the number of patches and devices. Note that the FID evaluation results are independent of the specific prompts used for the visual demonstrations.

# B Ablation Study

**Patch Number:** We analyze the impact of the setting of patch numbers $M$ on PipeFusion as shown in Figure 11. According to the analysis presented in Table 1, $M$ does not affect the communication cost. However, as shown in the figure, the lowest latency is achieved when $M$ is equal to the number of GPUs, $N$. If $M$ is large, the input size for operators becomes smaller, which can negatively impact the computational efficiency of the operator. In contrast, if $M$ is small, the efficiency of overlapping communication and computation decreases.

**Warmup Step**: The number of warmup steps can negatively impact the overall performance of PipeFusion. The input temporal redundancy is relatively small during the initial few diffusion timesteps. Therefore, it is usually to employ a few warmup steps to enable synchronous communication without resorting to stale activations, which would introduce waiting time, commonly referred to as bubbles, into the pipeline. As the number of sampling steps increases, the latency will also increase, as shown in Table 3. If the proportion of warmup steps is relatively small, no action is required. Our experiments involved a relatively small number of sampling steps, specifically 20 and 28. In other related work, it is common to use 50 steps [16] or even 100 steps [12], which helps to alleviate the overhead of warmup.

To mitigate the performance degradation caused by warmup, we can separate the warmup steps from the remaining working steps and allocate different computational resources to them. The output feature maps after the warmup steps can then be transmitted from the warmup devices to the

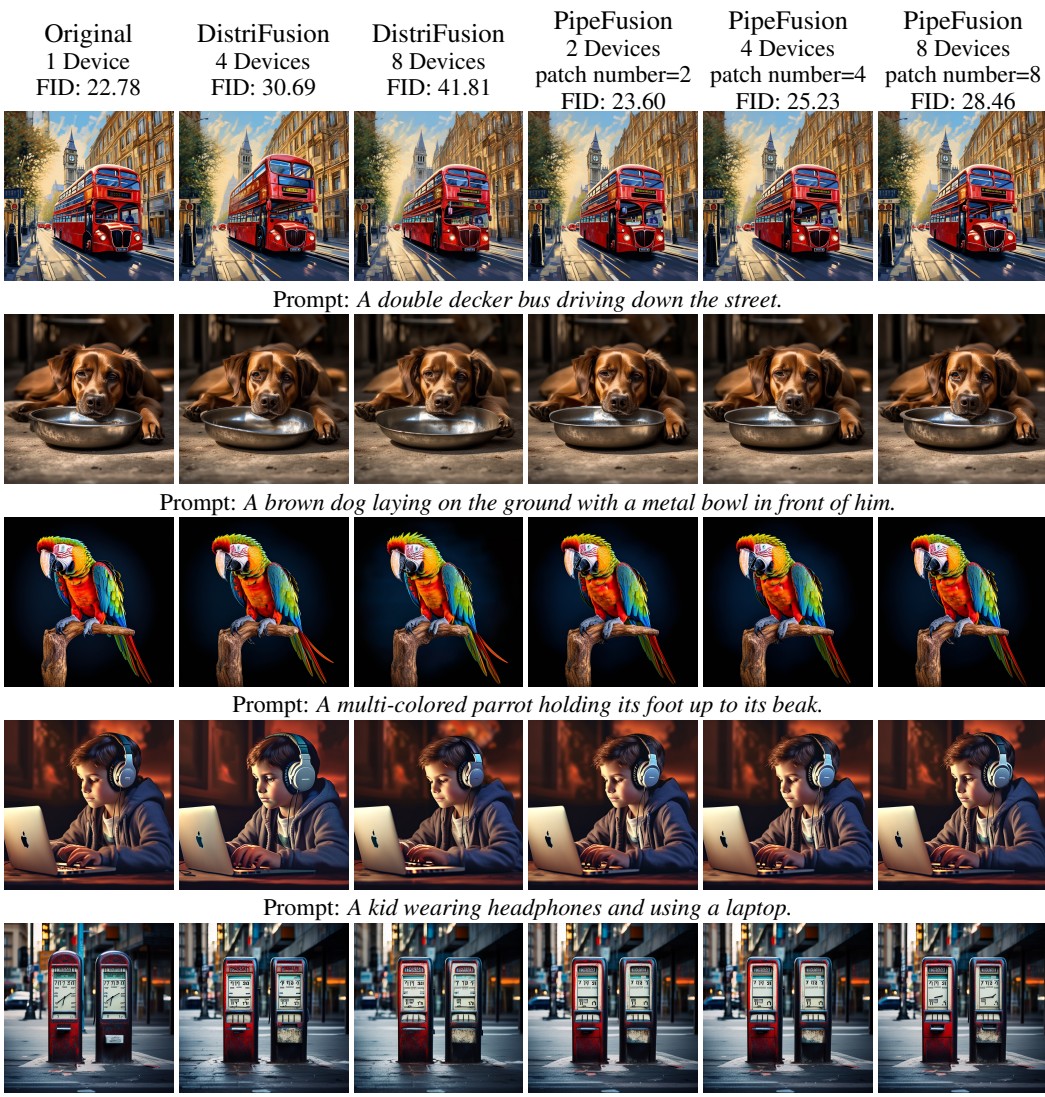

Figure 10: Above: Showcases for 1024px Generation Images of PipeFusion and DistriFusion using Pixart. Bottom: FID Scores Evaluated for Pixart and Flux.1. We use a 20-step DPM-Solver with the warmup step set to 1 for DistriFusion and PipeFusion. We use the COCO Captions 2014 [37] dataset to evaluate the FID scores. During the evaluation, a subset comprising 30,000 images is sampled from the validation set and resized to 256px to serve as the reference dataset. Concurrently, each experiment generates 30,000 images of 256px, each paired with a caption derived from the COCO Captions 2014 dataset, as the sample dataset. The quality of images generated by PipeFusion closely resembles that of the original images, regardless of whether 4 or 8 devices are used, and across varying patch numbers. FID above the images is computed against the ground-truth images using Clean-FID [38].

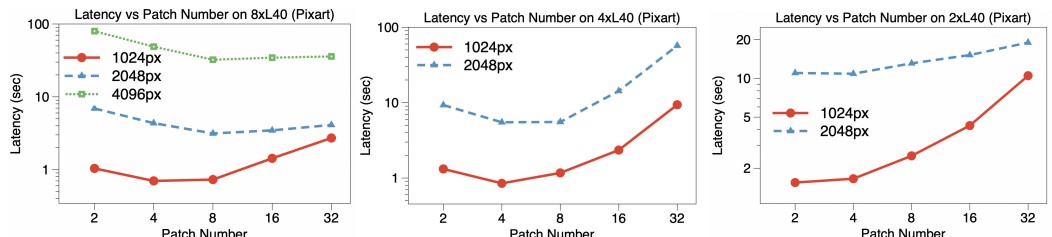

Figure 11: Latency of PipeFusion (without warmup) for various patch numbers $M$ on 2x, 4x, and 8xL40 GPUs in image generation tasks at three different resolutions using Pixart.

| Warmup | pixart | | sd3-medium | | Flux.1-dev |
|--------|--------|----------|------------|----------|-----------|
| | pp=8 | cfg=2, pp=8 | pp=8 | cfg=2, pp=8 | pp=8 |
| 0 | 0.71 | 0.66 | 1.05 | 0.83 | 5.48 |
| 1 | 0.82 (+15%) | 0.69 (+4%) | 1.16 (+10%) | 0.87 (+4%) | 5.53 (+1%) |
| 2 | 0.91 (+28%) | 0.70 (+6%) | 1.27 (+21%) | 0.92 (+11%) | 6.00 (+9%) |

Table 3: Impact of the warmup step number to Pixart, sd3-medium and Flux.1-dev on 1024px image generation, where pp indicates pipefusion (pp) parallel degree.

working devices. The warmup phase can apply sequence parallelism to fully utilize the computational resources.

We found that the setting of warmup steps is highly dependent on the model. By initializing the KV buffer to zero and not performing warmup, Pixart can still achieve satisfactory image generation results. Through dynamic detection of the difference between the latent space input and the previous diffusion step, we can automatically set the warmup steps. These optimizations for warm-up will be our future work.

| Method | 1024px | 2048px | 4096px |
|--------|--------|--------|--------|
| Tensor Parallel | 1.22 | 7.07 | 36.33 |
| DistriFusion | 0.77 | 2.48 | 25.24 |
| SP-Ring | 1.37 | 2.46 | 23.31 |
| SP-Ulysses | 2.59 | 3.54 | 27.41 |
| PipeFusion | **0.66** | **2.59** | **22.39** |

Table 4: PixArt latency (sec) on $8\times$A100 GPUs with NVLink interconnect. PipeFusion consistently delivers the lowest latency.

**Evaluation on NVLink connected GPUs:** To evaluate the sensitivity of PipeFusion to hardware interconnect bandwidth, we conducted additional experiments on $8\times$A100 GPUs connected via NVLink (high-bandwidth) in contrast to the default $8\times$L40 PCIe setting. Table 4 reports the latency for PixArt inference across different resolutions. PipeFusion consistently achieves the lowest latency across all resolutions and remains robust even under high-bandwidth NVLink settings. While the relative performance gap narrows at higher resolutions due to the increasing dominance of computation cost (i.e., the shift from communication- to computation-bound regimes), PipeFusion is still competitive or superior to all baselines. We also emphasize that the FID score is invariant to GPU type or interconnect bandwidth, as it is solely determined by the model architecture and inference algorithm.

