# OpenReview forum: "PipeFusion: Patch-level Pipeline Parallelism for Diffusion Transformers Inference"
_NeurIPS.cc/2025/Conference — NeurIPS 2025 poster_

### Official Review · Reviewer_9xjo · 2025-07-03

**Clarity:** 3
**Significance:** 3
**Originality:** 3
**Rating:** 5
**Confidence:** 5

**Summary:**

PipeFusion introduces a patch-level pipeline parallelism to accelerate inference for DiTs on multi-GPUs. It divides the input into multiple patches and maps the model layers across GPUs, enabling pipelined parallel processing. Also, PipeFusion leverages the high temporal redundancy between consecutive steps by reusing one-step stale feature maps to reduce the communication cost compared to existing parallelization methods such as tensor parallelism, sequence parallelism, and DistriFusion. Experiments on DiT models, including Pixart, Stable-Diffusion 3, and Flux.1, demonstrate substantial improvements in latency with minimal impact on image generation quality.

**Questions:**

In your experimental results, you note that for Pixart at 4096 px resolution, the performance gap between PipeFusion and SP narrows, and both methods exhibit similar scalability. While you attribute this to the exceptionally long sequence length making computation the dominant bottleneck rather than communication, the analysis remains relatively high-level.

Could you provide a more detailed breakdown or profiling of the memory cost and computational bottlenecks at this resolution?
Specifically:
1. Can you quantify how the memory footprint (for parameters, activations, and intermediate buffers) scales with resolution and sequence length for PipeFusion and SP?
2. Is there a point at which the communication-to-computation ratio fundamentally shifts, and can you provide empirical or theoretical evidence for where this occurs?
3. Have you profiled GPU utilization, memory bandwidth, or kernel efficiency at 4096px to pinpoint what limits further speedup for PipeFusion?

[Significance] A more granular analysis of these aspects would help clarify the practical and theoretical limits of your approach and guide future work.

**Ethical Concerns:**

["NO or VERY MINOR ethics concerns only"]

**Final Justification:**

The authors have provided a detailed breakdown of the performance, which is very helpful. My main concerns regarding the significance have been addressed.

**Limitations:**

Yes

**Quality:**

4

**Strengths And Weaknesses:**

Strengths
1. It’s a novel approach to combine both patch-level and layer-level partitioning for DiT inference. This is distinct from prior work that primarily focused on sequence or tensor parallelism.
2. The methods of this paper are easy to follow and adopt. The core concept can be simply handled by specifically writing the schedule in the compiler. The only additional thing to pay attention to is the input temporal redundancy.
3. Authors provide a thorough comparison with prior parallelization methods and clearly articulate the scenarios where PipeFusion is most beneficial, as well as its limitations.

Weaknesses:
1. As the discussion of stability section, when computation becomes dominant, the benefit of PipeFusion’s communication reduction is less pronounced, e.g., Pixart 4096px.

---

> ### Author Rebuttal · Authors · 2025-07-29
>
> We sincerely thank the reviewer for the insightful and constructive comments.
>
> We apologize for the misleading wording in the last paragraph of Sec. 4.1.1. Below, we provide clarifications and additional analysis to address your concerns:
>
> First, on the statement "PipeFusion and SP exhibit similar scalability at 4096px":
>
> We must point out that the term "similar" is imprecise. In fact, PipeFusion still demonstrates clear advantages at 4096px. As shown in Figure 5 (8×L40), PipeFusion achieves an end-to-end (E2E) latency of 32.1s, while SP’s best configuration (Ulysses=2, Ring=4) reaches 37.31s, yielding a 1.16× E2E speedup. Despite entering a compute-bound regime at 4096px, PipeFusion still outperforms SP by 1.16× end-to-end — a strong indication of its robustness even under diminishing communication relevance.
>
> Moreover, analyzing the communication component itself reveals a stark difference: PipeFusion communication share = (32.1 - 244.89/8) / 32.1 = 4.6%. SP communication share = (37.31 - 244.89/8) / 37.31 = 17.9%.This shows that PipeFusion significantly reduces communication overhead even at high resolutions. We will revise the wording and replot the relevant figure to reflect this difference better.
>
> Second, on the statement "This is likely due to the exceptionally long sequence, where the communication overhead becomes less significant, thereby diminishing the advantage of PipeFusion’s communication efficiency.":
>
> The analysis is vague and overly high-level. We therefore provide a more detailed analysis of how PipeFusion’s speedup varies across different input resolutions—1024px, 2048px, and 4096px.
>
> Fact 1: PipeFusion’s speedup for the communication component over SP (measured as SP comm. time / PipeFusion comm. time) becomes increasingly significant as resolution grows. For example, on 8×L40, at 1024px, PipeFusion reduces the latency of communication component to 54%(-46%); at 2048px, it is reduced to 24%(-76%); at 4096px, it is reduced to 22%(-78%). The PipeFusion’s speedup for the communication component becomes more pronounced at higher resolutions. This trend is attributable to longer sequences resulting in an increase in absolute communication time, and PipeFusion only benefits the communication component rather than the computing one.
>
> |                         | 1024px     | 2048px     | 4096px     |
> |-------------------------|------------|------------|------------|
> | **seqLen**              | 4,096      | 16,384     | 65,536     |
> | **8×L40 PipeFusion Latency (sec)** | 0.82       | 3.27       | 32.1       |
> | **8×L40 SP Latency (sec)**        | 1.22       | 5.07       | 37.3       |
> | **PipeFusion Comm. Time (sec)**   | 0.47       | 0.57       | 1.49    |
> | **PipeFusion Comm. Share**        | 57.3%      | 17.4%      | 4.6%       |
> | **SP Comm. Time (sec)**           | 0.87       | 2.37       | 6.69   |
> | **SP Comm. Share**                | 71%        | 47%        | 18%        |
> | **Comm. PipeFusion/SP**           | 54%        | 24%        | 22%        |
> | **Baseline (single GPU, sec)**    | 2.8        | 21.6       | 244.89     |
>
> Fact 2: The communication-to-computation ratio decreases as resolution increases. In PixArt, FLOPs are dominated by self-attention (which scales quadratically with sequence length). In contrast, communication cost (based on PipeFusion Table 1) grows only linearly with sequence length. This causes the communication-to-computation ratio to decrease — at 4096px, it is 0.1690 that of 1024px.
> As a result, although PipeFusion increasingly accelerates the communication component, this component constitutes a shrinking portion of total latency. For example, SP's communication share drops from 71% (1024px) to 47% (2048px) and 18% (4096px).
>
> | Input  | SeqLen  | Compute (GFLOPs) | PipeFusion Comm. (GB) | Comm/Compute Ratio (Normalized) |
> |--------|---------|------------------|------------------------|---------------------|
> | 1024px | 4,096   | 6,507.61         | 0.0175                 | 1.0000              |
> | 2048px | 16,384  | 51,952.85        | 0.0703                 | 0.5032              |
> | 4096px | 65,536  | 623,373.28       | 0.2816                 | 0.1680              |
>
> These two facts explain the observed behavior: PipeFusion yields stronger absolute gains on the communication part as sequence length increases, but the overall end-to-end speedup becomes less dramatic since communication accounts for a smaller share of total runtime. We will clarify this nuanced trade-off in the revised manuscript.
>
> **For Q2**:  While PipeFusion consistently accelerates the communication component, the relative speedup on end-to-end latency diminishes because computation grows quadratically, whereas communication scales linearly.
>
> However, we emphasize that this trade-off is tied to GPU scale. If GPU numbers scale proportionally with compute (e.g., 2 GPUs for 1024px, 8 for 2048px, 32 for 4096px), the communication share remains stable. In such cases, PipeFusion's advantage becomes more pronounced, making it highly effective in large-scale deployments.
>
> Third, on profiling memory, GPU utilization, and kernel efficiency at 4096px. Using DCGM profiling, we summarize our findings in the following table:
>
> | Metric                          | 1024px   | 2048px   | 4096px   |
> |--------------------------------|----------|----------|----------|
> | Param mem. (GB)                | 0.15     | 0.15     | 0.15     |
> | Text encoder mem. (GB)         | 10.75    | 10.75    | 10.75    |
> | Stale KV buffer mem. (GB)      | 0.03075  | 0.123    | 0.492    |
> | Activations mem. (GB)          | 0.91925  | 1.047    | 1.578    |
> | Overall mem. (GB)              | 11.85    | 12.07    | 12.97    |
> | Latency (Sec)                  | 0.82     | 3.27     | 32.1     |
> | Mem Bandwidth Util.            | 5%       | 4%       | 2%       |
> | GPU SM Activity                | 37%      | 65%      | 97%      |
> | PipeFusion Comm / Latency      | 0.57     | 0.17     | 0.046    |
> | SP Comm / Latency              | 0.71     | 0.46     | 0.178    |
>
> **For Q1**: As shown in the above table, PipeFusion’s activation and stale-KV memory footprint remains low and does not pose a scaling bottleneck.
>
> **For Q3**: On 4096px case, the above table shows that GPU SM activity reaches 97%, while memory bandwidth utilization is only 2%. This indicates that the model is compute-bound, not memory-bound, at this resolution. Consequently, further speedup would require optimizing kernel execution, such as through improved CUDA kernels or quantization techniques. These compute-level optimizations are orthogonal to PipeFusion’s communication strategy and can be applied in combination to further enhance performance.
> In summary, we appreciate your comments, which led us to refine both the empirical explanation and theoretical understanding. We will revise the manuscript accordingly to ensure the significance of PipeFusion is not underestimated.

---

> > ### Comment · Reviewer_9xjo · 2025-08-05
> >
> > Thank you for your informative response, which addresses my main concerns. I am comfortable raising my rating to 5.

---

> > > ### Author Response · Authors · 2025-08-05
> > > **Thanks!**
> > >
> > > Thank you very much for your thoughtful engagement and for considering our clarifications. We sincerely appreciate your updated rating and are grateful for your support.

---

### Official Review · Reviewer_7h7H · 2025-07-03

**Clarity:** 3
**Significance:** 3
**Originality:** 3
**Rating:** 5
**Confidence:** 3

**Summary:**

The paper presents a novel parallelization strategy, dubbed PipeFusion, for efficient inference of Diffusion Transformer. The method partitions both input images and model layers across multiple GPUs, employing patch-level pipeline parallelism to mitigate communication overhead and leverage temporal redundancy. The approach is validated across various state-of-the-art diffusion Transformer models, demonstrating its superior performance in terms of latency, scalability and image quality.

**Questions:**

Please see the above **weaknesses** section.

**Ethical Concerns:**

["NO or VERY MINOR ethics concerns only"]

**Final Justification:**

My concerns are resolved in the rebuttal.

**Limitations:**

Yes

**Quality:**

3

**Strengths And Weaknesses:**

### Strengths

1. The proposed patch-level pipeline parallelism is an innovative and effective approach.

2. The experiment is extensive, covering multiple strong baselines incl. DistriFusion across various Diffusion Transformer architectures incl. Pixart, Stable-Diffusion.

3. Experimental results consistently favor PipeFusion across different image resolutions and model scales.

### Weaknesses

1. The proposed framework is only tested under one system configuration: 8 x L40 GPUs. As the pipeline stages and GPU to GPU communication bandwidth could greatly impact the performance (FID score and throughput) of the proposed method. The authors are suggested to test under different system settings (e.g., communication bandwidth).

2. It would be great to have more studies regarding the performance impact of stale representations, as this is one important reason of why the proposed PipeFusion has lower FID compared to DistriFusion, potentially by testing the FID score under different level of staleness.

---

> ### Author Rebuttal · Authors · 2025-07-29
>
> Q1:1. The proposed framework is only tested under one system configuration: 8 x L40 GPUs. As the pipeline stages and GPU to GPU communication bandwidth could greatly impact the performance (FID score and throughput) of the proposed method. The authors are suggested to test under different system settings (e.g., communication bandwidth).
>
> Thank you for your valuable feedback. We fully agree that system configuration, especially inter-GPU communication bandwidth, plays a critical role in the performance of parallel inference methods.
>
> To address this concern, we have conducted additional experiments on an alternative hardware platform: 8×A100 GPUs connected via NVLink, which provides significantly higher communication bandwidth than PCIe. The results are summarized in the table below for the Pixart model:
>
> | Latency (sec)      | 1024px | 2048px | 4096px |
> |----------------|--------|--------|--------|
> | tensor parallel| 1.22   | 7.07   | 36.33  |
> | distrifusion   | 0.77   | 2.476  | 25.24  |
> | sp-ring        | 1.37   | 2.46   | 23.31  |
> | sp-ulysses     | 2.59   | 3.54   | 27.41  |
> | **pipefusion** | **0.66** | **2.59** | **22.39** |
>
> As shown above, PipeFusion consistently achieves the lowest latency across all resolutions and remains robust even under high-bandwidth NVLink settings. While the relative performance gap narrows at higher resolutions due to the increasing dominance of computation cost (i.e., the shift from communication- to computation-bound regimes), PipeFusion remains competitive or superior across all configurations.
>
> We would also like to clarify that the FID score is determined solely by the model architecture and inference algorithm, and remains consistent across different hardware settings as long as the number of GPUs and the generated outputs are the same. Therefore, changes in GPU type or interconnect bandwidth do not affect FID evaluation.
>
> Q2: It would be great to have more studies regarding the performance impact of stale representations, as this is one important reason of why the proposed PipeFusion has lower FID compared to DistriFusion, potentially by testing the FID score under different level of staleness.
>
> Thank you for highlighting the importance of evaluating the impact of stale representations on diffusion performance. We plan to conduct controlled experiments that vary the level of activation staleness (e.g., by adjusting the freshness ratio or delaying KV usage across more steps), and measure the corresponding FID. This would help isolate the direct influence of stale feature reuse on output quality.
>
> Some works in our reference have begun exploring this space. For instance:
>
> - ∆-DiT [24] studies caching strategies for different DiT layers depending on the sampling stage;
>
> - Learning-to-Cache [13] analyzes layer-wise temporal redundancy and proposes selective reuse policies;
>
> - FrDiff [11] and DeepCache [12] observe that reusing high-level activations introduces minimal degradation while significantly accelerating inference.
>
> We anticipate combining PipeFusion’s low-communication pipelined inference with more stale steps offers a two‑axis acceleration: reducing both inter-device communication and redundant per-timestep computation, with minimal FID degradation. We will add this point to the Discussion section.

---

> ### Comment · Reviewer_7h7H · 2025-08-04
>
> I appreciate the authors’ informative response, which has addressed all of my concerns. Accordingly, I have increased the score to 5.

---

> > ### Author Response · Authors · 2025-08-04
> > **Thanks!**
> >
> > Thank you very much for your kind follow-up and for increasing the score to 5. We're glad that our response helped address your concerns. We sincerely appreciate your thoughtful feedback and constructive engagement throughout the review process.

---

### Official Review · Reviewer_2Wj8 · 2025-07-04

**Clarity:** 2
**Significance:** 2
**Originality:** 2
**Rating:** 4
**Confidence:** 3

**Summary:**

The paper proposes a parallelization scheme for diffusion model inference and demonstrates favorable performance on a range of models on 8 NVIDIA L40 GPUs. The primary consideration is to minimize latency caused by GPU-to-GPU communication. First, it leverages pipeline parallelism so that the communication is only required for passing the last activations in one device to the other. Second, it performs patch-wise processing and uses the cached activations from the previous denoising time step, leveraging the temporal similarity of activations across neighboring time steps in diffusion model inference. The results demonstrate an efficient inference pipeline that outperforms competing implementations.

**Questions:**

Q1: Would it be feasible to extend the proposed method to handle DiT-based video model inference?

Q2: Have you considered compressing the activations to further reduce communication cost?

Q3: Is the DiT activation as smooth as the UNet activations across different denoising time steps.

**Ethical Concerns:**

["NO or VERY MINOR ethics concerns only"]

**Final Justification:**

After checking the rebuttal and the reviews from the other reviewers, I lean towards keeping my original rating. I feel the merit provided in the paper warrants a publication.

**Limitations:**

Yes

**Quality:**

2

**Strengths And Weaknesses:**

Strengths
- Achieve clear speed gain over competing methods
- Have a lower memory footprint.
- I think the proposed method shows a new direction for diffusion model inference optimization. I am unaware of prior works leveraging pipeline parallelism for DiT acceleration.

Weakness
- A nice integration of known techniques. However, the added value is a bit limited.

---

> ### Author Rebuttal · Authors · 2025-07-29
>
> Q1: Would it be feasible to extend the proposed method to handle DiT-based video model inference?
>
> Thank you for the insightful question. We believe PipeFusion is highly compatible with DiT-based video models and can naturally extend to this domain with minimal adaptation.
>
> From a model architecture perspective, PipeFusion already supports Multimodal DiT (MM-DiT) structures, which inject both text and image conditions through a shared unified attention mechanism, enabling joint cross-modal interaction within the same transformer layers. As demonstrated in our experiments with Flux.1 and Stable Diffusion 3, both of which are MM-DiT structures. Notably, the most recently released video generation models—such as Wan2.1, HunyuanVideo, Mochi-1, and CogVideo—adopt similar MM-DiT backbones. PipeFusion can support these models seamlessly.
>
> From a tensor shape perspective, both image and video DiTs operate over latent tensors with the shape (batch_size, sequence_length, hidden_size). While image models flatten spatial dimensions (height × width) into sequence tokens, video models flatten temporal and spatial dimensions (frames × height × width) into sequence tokens. Since PipeFusion performs pipeline parallelism along the sequence dimension, this difference does not affect its applicability, making video and image DiTs equally compatible.
>
> Moreover, from a performance standpoint, PipeFusion is particularly well-suited for video models with longer sequence lengths and larger models. Compared to sequence parallelism (SP), PipeFusion incurs significantly lower memory footprint by partitioning model parameters layer-wise across GPUs—an important advantage for video models with more parameters (e.g., StepVideo, ~30B). Additionally, video models typically exhibit longer sequence lengths, amplifying the communication burden for SP. As shown in the table below, our results already demonstrate that for Pixart at 1024px (sequence length 4K), PipeFusion achieves 1.85× communication speedup over SP, and at 4096px (sequence length 65K), the gain increases to 4.49×. We expect even greater benefits in video settings (e.g., Wan2.1 with 1080×720×5s input seqlen is 75,600 tokens).
> PS: Comm. Time is calculated by (8 × L40 latency - 1 × L40 baseline) / 8
>
> |                          | 1024px        | 2048px       | 4096px        |
> |--------------------------|---------------|--------------|---------------|
> | **seqlen**               | 4,096         | 16,384       | 65,536        |
> | **8×L40 PipeFusion Latency (sec)** | 0.82          | 3.27         | 32.1          |
> | **8×L40 SP Latency (sec)**        | 1.22          | 5.07         | 37.3          |
> | **PipeFusion Comm Time (sec)**    | 0.47          | 0.57         | 1.48875       |
> | **SP Comm Time (sec)**            | 0.87          | 2.37         | 6.68875       |
> | **Comm. PipeFusion / SP**         | 185%          | 416%         | 449%          |
> | **1×L40 Baseline (sec)**          | 2.8           | 21.6         | 244.89        |
>
> We will add a brief discussion on this extension in the paper and plan to include detailed benchmarks on video models in future work.
>
> Q2: Have you considered compressing the activations to further reduce communication cost?
>
> Thank you for the insightful suggestion. We agree that compressing activations is a promising direction for further reducing communication overhead in multi-GPU inference.
>
> As you noted, communication remains a significant component of end-to-end latency in certain scenarios. For instance, in our experiments on Pixart 1024px using 8×L40 GPUs, communication accounts for approximately 47% of the total inference latency (As shown in the above Table). In such cases, activation compression techniques—such as quantization or sparsification—could indeed further reduce bandwidth usage and improve performance. Work, such as SQ-DM, CompactFusion, and TDQ, has demonstrated the potential of such methods.
>
> SQ-DM: Fan Z, Dai S, Venkatesan R, et al. SQ-DM: Accelerating Diffusion Models with Aggressive Quantization and Temporal Sparsity[J]. arXiv preprint arXiv:2501.15448, 2025.
>
> TDQ: CompactFusion: So J, Lee J, Ahn D, et al. Temporal dynamic quantization for diffusion models[J]. Advances in neural information processing systems, 2023, 36: 48686-48698.
>
> CompactFusion: Luo J, Xiao Y, Xu J, et al. Accelerating Parallel Diffusion Model Serving with Residual Compression[J]. arXiv preprint arXiv:2507.17511, 2025.
>
> However, we also observed that the benefit of activation compression may diminish for high-resolution inputs. For example, in our 4096px Pixart-8×L40 setup, communication only contributes to 4.6% of the total latency. The program becomes compute-bound, and introducing additional operations (e.g., for compression or decompression) may negate the communication savings and even hurt performance.
>
> We will add a discussion on activation compression and its trade-offs to the main paper.
>
> Q3: Is the DiT activation as smooth as the UNet activations across different denoising time steps?
>
> The Δ‑DiT [24] paper highlights that both U-Net and DiT both exhibit high activation similarity between adjacent denoising steps. Empirical evidence (e.g., in DiTFastAttn [14], Learning‑to‑Cach [13]) also suggests that temporal redundancy is significant enough to be exploited for acceleration or compression.
> We have added a brief discussion of this point in the revised version.
>
> Chen P, Shen M, Ye P, et al. *Δ‑DiT: A Training-Free Acceleration Method Tailored for Diffusion Transformers*. arXiv preprint arXiv:2406.01125, 2024.
>
> Yuan Z, Zhang H, Pu L, et al. *DiTFastAttn: Attention Compression for Diffusion Transformer Models*. *Advances in Neural Information Processing Systems*, 2024, 37: 1196–1219.
>
> Ma X, Fang G, Bi Mi M, et al. *Learning-to-Cache: Accelerating Diffusion Transformer via Layer Caching*. *Advances in Neural Information Processing Systems*, 2024, 37: 133282–133304.

---

### Decision · Program_Chairs · 2025-09-17

**Decision:**

Accept (poster)

**Comment:**

The paper received all positive reviews, leading to a final acceptance recommendation.